# *OsABT* Is Involved in Abscisic Acid Signaling Pathway and Salt Tolerance of Roots at the Rice Seedling Stage

**DOI:** 10.3390/ijms231810656

**Published:** 2022-09-13

**Authors:** Danni Wen, Lingran Bao, Xuanzhu Huang, Xueduo Qian, Eryong Chen, Bo Shen

**Affiliations:** 1College of Life and Environmental Sciences, Hangzhou Normal University, Hangzhou 311121, China; 2Life School of Science and Technology, Henan Institute of Science and Technology, Xinxiang 453003, China

**Keywords:** rice, root, salt stress, WD40 proteins, ABA, Na^+^ and K^+^, ion toxicity, transcriptome analysis

## Abstract

Rice is a staple cereal crop worldwide, and increasing its yields is vital to ensuring global food security. Salinity is a major factor that affects rice yield. Therefore, it is necessary to investigate salt tolerance mechanisms in rice. Proteins containing WD40 repeats play important roles in eukaryotic development and environmental adaptation. Here, we showed that overexpression of *OsABT*, a gene encoding a WD40-repeat protein, enhanced salt tolerance in rice seedlings by regulating root activity, relative conductivity, malondialdehyde and H_2_O_2_ content, and O_2_^•−^ production rate. Root ion concentrations indicated that *OsABT* overexpression lines could maintain lower Na^+^ and higher K^+^/Na^+^ ratios and upregulated expression of salt-related genes *OsSOS1* and *OsHAK5* compared with the wild-type (WT) Nipponbare plants. Furthermore, Overexpression of *OsABT* decreased the abscisic acid (ABA) content, while downregulating the ABA synthesis genes *OsNCED3* and *OsNCED4* and upregulating the ABA catabolic gene *OsABA8ox2*. The yeast two-hybrid and bimolecular fluorescence complementation analyses showed that OsABT interacted with the ABA receptor proteins OsPYL4, OsPYL10, and PP2C phosphatase OsABIL2. A transcriptome analysis revealed that the differentially expressed genes between *OsABT* overexpression lines and WT plants were enriched in plant hormone signal transduction, including ABA signaling pathway under salt stress. Thus, *OsABT* can improve the salt tolerance in rice seedling roots by inhibiting reactive oxygen species accumulation, thereby regulating the intracellular Na^+^/K^+^ balance, ABA content, and ABA signaling pathway.

## 1. Introduction

Abiotic stresses, such as salt stress, drought, and extreme temperatures, seriously affect the growth and productivity of rice [1]. Salinity stress is one of the major constraints to rice growth, affecting over 20% of all arable land globally [2]. As an essential organ for rice growth and development, roots not only anchor plants and take up water and nutrients from the soil but also synthesize many substances [3]. Roots are the primary site for salt stress signal perception and can act as an early warning system [4].

Salt stress can cause ion toxicity in plants and interfere with K^+^ and Na^+^ homeostasis in the cytoplasm, thereby damaging root cell membrane selectivity [5]. Plants use the salt overly sensitive (SOS) pathway for salt stress signaling and development of Na^+^ tolerance [6,7]. OsSOS1 mediates Na^+^ influx and Na^+^ redistribution at the xylem parenchyma boundary and regulates Na^+^ efflux in the roots [8]. Vacuolar compartmentalization of Na^+^ can also reduce its accumulation in the cytoplasm, where the vacuolar Na^+^/H^+^ antiporter OsNHXs play an important role in regulating rice salt tolerance [9,10,11,12]. Potassium plays a key role in regulating plant growth and cellular salt tolerance [13]. Several genes encoding root K^+^ uptake channels have been identified in rice. OsAKT1, OsKAT1, OsHAK1, OsHAK5, and OsHAK21 participate in salt stress responses by increasing K^+^ uptake and maintaining a low Na^+^/K^+^ ratio [14,15,16,17,18].

Reactive oxygen species (ROS) have been identified as important signaling molecules that regulate root development, cell differentiation, elongation, and hair formation as well as lateral root emergence [19,20,21,22]. However, a large number of intracellular ROS are produced under stress, causing oxidative damage to proteins, RNA, and DNA molecules. Therefore, the ability to scavenge ROS is of great significance in improving stress tolerance [23]. It was reported that a drought-sensitive mutant 1 (*dsm1*) regulates the response to drought stress by regulating ROS scavenging in rice [24]. *OsVTC1-3* RNAi rice accumulated a large amount of O_2_^•−^ and H_2_O_2_ in the roots under salt stress, and the tolerance to salt stress was reduced [25]. Overexpression of *OsMADS25* can reduce the level of ROS in rice roots by enhancing the activity of antioxidant enzymes, regulating root growth, and conferring salt tolerance [26]. *OsR3L1* maintains ROS homeostasis under salt stress by regulating the activities of ROS-scavenging enzymes in the early stages of root development, thereby improving salt tolerance [27]. These results show that ROS play an important role in the growth and development of rice roots.

Abscisic acid (ABA) plays a key role in plant development by regulating a series of physiological processes, such as seed germination and dormancy, root development, stomatal regulation, and abiotic stress resistance [28]. ABA is biosynthesized in the root phloem and transported to the leaves and other tissues to perform its function [29]. ABA accumulates significantly under osmotic stress conditions such as drought and salinity [30,31,32]. Under salt stress conditions, ABA binds to receptors that regulate the expression of ABA-responsive genes through the ABA signaling pathway [33,34,35,36,37] that has three core components: PYR/PYL/RCAR ABA receptors (PYLs), type 2C protein phosphatases (PP2Cs), and SNF1-related protein kinase 2 (SnRK2s) [38]. In recent years, some candidate genes related to the ABA signaling pathway have been identified that play a key role in promoting root stress response in rice [39,40,41,42,43,44].

WD40 proteins are extremely abundant and highly conserved in eukaryotes [45]. They act as important flexible scaffolds that mediate protein–protein or protein-DNA interactions and play pivotal roles in diverse cellular processes, including RNA processing, immune responses, signal transduction, gene regulation, microtubule organization, and hormone response [46,47,48,49]. The ABA signaling terminator gene (*ABT*), which encodes a WD40-repeat protein in *Arabidopsis*, is critical for seed germination and seedling establishment [50]. In our previous studies, we identified *OsABT* (*Os03g0738700*) and found that its overexpression significantly enhanced salt tolerance in rice seedlings [51]. However, the mechanism underlying salt tolerance mediated by *OsABT* remains unknown. In this study, we demonstrate that *OsABT* can improve the salt tolerance of rice roots at the seedling stage by inhibiting ROS accumulation, maintaining intracellular Na^+^ and K^+^ balance, and regulating ABA content. Meanwhile, the OsABT protein interacts with OsABIL2, OsPYL4, and OsPYL10 to mediate root salt tolerance by affecting the ABA signaling pathway and the expression of ABA-related genes at the rice seedling stage.

## 2. Results

### 2.1. Identification of OsABT Overexpression Rice

To identify the functions of *OsABT*, we transformed it into wild-type (WT) Nipponbare plants and identified two-week-old T_6_ generation transgenic rice OsABT–OE lines (OE–3 and OE–4) using reverse transcription-polymerase chain reaction (RT-PCR) and quantitative real-time polymerase chain reaction (qRT-PCR) (Appendix A). The transgenic rice lines OE–3 and OE–4 showed significantly higher *OsABT* expression of roots than the WT plants and thus could be deemed as *OsABT* overexpression lines, which were used for subsequent experiments.

### 2.2. Overexpression of OsABT Enhances the Salt Tolerance of Rice Seedlings

To further investigate the salt tolerance mechanism mediated by *OsABT*, the OsABT–OE lines (OE–3 and OE–4) and WT plants were exposed to 200 mM NaCl. Before salt treatment, the OsABT–OE and WT phenotypes did not significantly differ from each other (Figure 1A). After salt treatment for 12 and 24 h, the OsABT–OE lines showed significantly less leaf curling and higher relative leaf water content than the WT plants (Figure 1B–D). These results indicate that the *OsABT* overexpression lines had a better salt-tolerant phenotype at the seedling stage.

The two-week-old rice seedlings were in a rapid growth period, and their root activity gradually increased. Salt treatment inhibited this increase in root activity. Before salt treatment, no significant difference in root activity was observed between the OsABT–OE lines and WT plants. At 24 h of salt treatment, the root activity of OsABT–OE lines was significantly higher than that of WT plants (Figure 1E). The relative conductivity of roots and leaves increased gradually with the extension of salt treatment time. Under salt stress, the relative conductivity of the roots and leaves of the OsABT–OE lines was significantly lower than that of WT, especially at 24 h of salt treatment (Figure 1F,G). The malondialdehyde (MDA) content in the roots and leaves of the OsABT–OE lines was also lower than that of the WT plants (Figure 1H,I). These results suggest that *OsABT* can improve the salt tolerance of rice seedlings by increasing root activity and reducing relative conductivity and MDA content.

Salt stress leads to ROS accumulation that adversely affects plant tissues [23]. Although the O_2_^•−^ production rate and H_2_O_2_ content in rice roots increased in all plant lines following 12-h and 24-h salt treatments, these indicators were significantly lower in OsABT–OE lines than in WT plants at the same time of salt treatment (Figure 2). These results show that the *OsABT* overexpression lines may enhance the salt tolerance of rice by inhibiting O_2_^•−^ production and reducing H_2_O_2_ content in roots, indicating that *OsABT* has an inhibitory effect on ROS accumulation under salt stress.

### 2.3. Overexpression of OsABT Regulates Na^+^ and K^+^ Levels under Salt Stress

An important aspect of salt tolerance is the avoidance of Na^+^ accumulation and K^+^ homeostasis [52]. Two-week-old hydroponically grown OsABT–OE lines and WT plants were exposed to 200 mM NaCl. The level of Na^+^ in the roots increased continuously in all plant lines under salt stress. However, the OsABT–OE lines accumulated significantly less Na^+^ than the WT plants, especially after 24 h of salt treatment (Figure 3A). In contrast to Na^+^ levels, K^+^ levels declined in the roots of all plant lines during salt stress. At 24 h of salt treatment, the K^+^ content in OsABT–OE lines was higher than that in WT plants (Figure 3B). A decrease in the K^+^/Na^+^ ratio was detected in all plant lines under salt stress. However, a markedly higher K^+^/Na^+^ ratio was observed in the OsABT–OE lines than in the WT plants after 24 h of salt treatment (Figure 3C). These results imply that *OsABT* can inhibit Na^+^ accumulation and maintain a high K^+^/Na^+^ ratio in the roots, thereby maintaining the balance between Na^+^ and K^+^ in rice root cells.

Next, we determined the root expression of *OsSOS1* and *OsHAK5* by qRT-PCR. Before salt treatment, the expression of *OsSOS1* and *OsHAK5* was not significantly different between the OsABT–OE line (OE–3) and WT plants. Salt stress significantly induced the expression of these genes in all plant lines; however, this tendency was enhanced in OE–3 plants. After 24 h of salt treatment, the root expression levels of *OsSOS1* and *OsHAK5* in OE–3 were 1.42- and 5.41-fold of that in WT, respectively (Figure 3D,E). These results reveal that overexpression of *OsABT* regulates Na^+^ and K^+^ levels in rice roots under salt stress by enhancing the expression of *OsSOS1* and *OsHAK5* genes.

### 2.4. Overexpression of OsABT Inhibits ABA Synthesis under Salt Stress

Abscisic acid (ABA) greatly contributes to plant growth and development and plays a key role in controlling the adaptive response of plants to environmental stresses. ABA accumulation and biosynthesis genes are upregulated under salt stress [53]. We found that the ABA content in the roots of the OsABT–OE line (OE–3) was lower than that of the WT plants. Meanwhile, the leaf ABA content showed first an increasing and then a decreasing trend in all plant lines under salt stress, but this tendency was weakened in OE–3 (Figure 4A,B).

The root expression of the ABA synthesis genes *OsNCED3* and *OsNCED4*, and the ABA catabolic gene *OsABA8ox2* were further examined. The results showed that the expression of these genes was dramatically induced in all plant lines under salt stress. Compared to the WT plants, the expression of *OsNCED3* and *OsNCED4* in OE–3 was lower, while that of *OsABA8ox2* in OE–3 was higher. At 24 h of salt treatment, the expression levels of *OsNCED3* and *OsNCED4* in OE–3 was only 43.55% and 46.24% of WT, while the expression of *OsABA8ox2* was 1.59-fold of WT (Figure 4C–E). ABA is synthesized in roots and transported to the leaves for accumulation under salt stress [33]. These results indicate that *OsABT* can inhibit ABA synthesis in rice roots by inhibiting the expression of ABA synthesis genes and promoting the expression of ABA catabolic genes, thereby reducing the leaf ABA content.

### 2.5. OsABT Interacts with Key Factors of the ABA Signaling Pathway

The ABA signaling pathway is regulated by three major components: PYR/PYL/RCAR ABA receptors (PYLs), type 2C protein phosphatases (PP2Cs), and SNF1-related protein kinase 2 (SnRK2s) [38]. Based on the ABA content and ABA biosynthetic gene expression results, we speculated that *OsABT* likely regulates the ABA signaling pathway. Therefore, we performed a yeast two-hybrid (Y2H) analysis to identify the interaction between OsABT and the ABA receptors OsPYL4, OsPYL10, and the PP2C protein OsABIL2. In the SD/-Leu/-Trp/-His/-Ura medium, OsABT fused to the GAL4-binding domain (BD) interacted with OsABIL2, OsPYL4, and OsPYL10 fused to the GAL4 activation domain (AD) in yeast cells (Figure 5A).

To further confirm the protein-protein interaction between OsABT and OsABIL2, OsPYL4, and OsPYL10 in plant cells, bimolecular fluorescence complementation (BiFC) assays were used. The C- and N-terminal regions of the yellow fluorescent protein (YFP) were coupled to OsABT and the three ABA signaling proteins OsABIL2, OsPYL4, and OsPYL10. These constructs were co-induced into the leaf cells of Nicotiana benthamiana. A reconstituted YFP signal was observed in both cytoplasm and nucleus, whereas no YFP signal was observed in the negative control cells harboring an empty vector (Figure 5B, Appendix A). These data indicate that OsABT physically interacts with OsABIL2, OsPYL4, and OsPYL10 in plant cells.

### 2.6. Transcriptomic Analysis of OsABT Overexpression Rice Roots under Salt Stress

To elucidate the molecular network underlying *OsABT*-regulated salt tolerance, we examined changes in gene expression in the roots of a two-week-old OsABT–OE line (OE–3) and WT plants using transcriptome deep sequencing (RNA-seq). We identified 1950 differentially expressed genes (DEGs; 954 upregulated and 996 downregulated) in OE–3 plants compared to the WT before salt treatment and 3499 DEGs (2955 upregulated and 544 downregulated) after 24 h of salt treatment (Appendix A). Considering that OsABT interacts with key factors in the ABA signaling pathway, we analyzed genes associated with plant hormone signal transduction, including ABA signal transduction, in the transcriptome data obtained for plants under salt stress (Appendix A).

We found that 13 DEGs in the ABA signaling core were regulated by *OsABT* after 24 h of salt treatment. Compared with WT, nine DEGs related to the ABA signal transduction pathway were downregulated in the roots of the OsABT–OE line, which included ABA receptors *OsPYL4*, *OsPYL9*, and *OsPYL5*; PP2C family proteins *OsPP108* and *OsABIL2*; SnRK2 family proteins *OsSAPK1*, *OsSAPK2*, and *OsSAPK9*; and bZIP transcription factor *OsABI5* (Figure 6). In contrast to the expression of ABA signal transduction genes, most DEGs in the other seven plant hormone signal transduction pathways were upregulated in the roots of the OsABT–OE line (Appendix A). These results suggest that *OsABT* negatively regulates the ABA signal transduction pathway.

To confirm these results, we detected the expression of DEGs using qRT-PCR. The expression of the ABA signal transduction-related genes *OsPYL5*, *OsPYL9*, and *OsABIL2*; ABA-responsive genes *OsLEA3* and *OsRAB16A* was significantly induced by salt stress. Compared to the WT roots, the OE–3 roots had a lower expression of these genes. At 24 h of salt treatment, the expression of the ABA-responsive genes *OsLEA3* and *OsRAB16* decreased by 54.10% and 29.58% compared with the WT plant, while ABA signal transduction-related genes *OsPYL5*, *OsPYL9*, and *OsABIL2* decreased by 26.38%, 28.01%, and 20.41%, respectively. These results are consistent with the transcriptome data and show that *OsABT* negatively regulates the expression of ABA-related genes.

## 3. Discussion

The root is a vital organ for absorbing water and other nutrients. More importantly, several studies have reported that modulating root development improves stress tolerance and increases yield in crops [3,4,54]. In this study, we found that *OsABT* overexpression lines showed an obvious and stable salt-tolerant phenotype compared to WT under 200 mM NaCl treatment (Figure 1A–C). At the same time, the relative conductivity, MDA content, O_2_^•−^ production rate, and H_2_O_2_ content in the roots of *OsABT* overexpression lines were lower than those of WT (Figure 1F–I and Figure 2). It is well known that ROS homeostasis is related to oxidative stress. At low cellular concentrations, ROS act as signaling molecules. Under severe biotic and abiotic stresses, ROS overproduction leads to impaired cell growth [23]. Therefore, plants have evolved complex systems to regulate ROS homeostasis and protect themselves from oxidative stress [24,25,26,27]. *OsABT* can enhance salt tolerance in rice by inhibiting the production of O_2_^•−^ and H_2_O_2_ in the roots of rice seedlings.

Salt stress can cause cytoplasmic Na^+^ accumulation and cell damage in rice. In addition, it causes reduction in the cellular K^+^ content, changes in the intracellular membrane potential, and hyperpolarization of the cell membrane. Therefore, maintaining the balance between Na^+^ and K^+^ in cells is essential for rice growth under salt stress [55]. Studies have shown that salt stress can lead to an increase in the Na^+^ content and a decrease in the K^+^ content of the roots and leaves [56], consistent with the changes in Na^+^ and K^+^ concentrations under salt stress in this study. Meanwhile, the roots of *OsABT* overexpression lines had lower Na^+^ content, lower K^+^ content, and higher K^+^/Na^+^ ratios than those of WT (Figure 3A–C). Furthermore, the root expression of *OsSOS1* and *OsHAK5* in *OsABT* overexpression lines was significantly upregulated compared with the WT plants under salt stress (Figure 3D,E), indicating that *OsABT* may regulate the expression of *OsSOS1* and *OsHAK5* to reduce the accumulation of Na^+^ and maintain the balance of Na^+^ and K^+^ in root cells under salt stress, thereby improving salt tolerance in rice.

Plant hormones, particularly ABA, can control gene expression through signal transduction and increase plant stress resistance [28]. Huang et al. [57] demonstrated that 9-cis-epoxycarotenoid dioxygenase (NCED) is a rate-limiting enzyme in ABA biosynthesis. *OsABA8ox* encodes ABA 8′-hydroxylase that catalyzes the committed step of ABA catabolism [58]. In this study, ABA content in the roots and leaves of *OsABT* overexpression lines was lower than that in WT plants (Figure 4A,B). Meanwhile, the root expression of *OsNCED3* and *OsNCED4* in the *OsABT* overexpression lines was lower than that in the WT plants under salt treatment, whereas the ABA catabolic gene *OsABA8ox2* had the opposite effect (Figure 4C–E). It is well known that ABA is biosynthesized in the root phloem and transported to the leaves and other tissues to function [33], showing that *OsABT* regulates the ABA content by inhibiting the expression of ABA synthesis genes and promoting the expression of ABA catabolic genes in rice roots, thereby affecting the accumulation of ABA in leaves.

As scaffold protein, proteins containing WD40 repeats may function as a component of protein complexes [47]. The WD40 protein XIW1 is a nucleocytoplasmic shuttle protein and plays an active role in ABA responses by interacting with and maintaining stability of ABI5 in the nucleus [59]. OsRACK1A (a WD40 type protein) can interacted with many salt-stress suppressed proteins directly [52]. ABT is a protein containing seven WD40 repeats in Arabidopsis. Wang et al. [50] found ABT interacts with PYR1/PYL/RCAR (PYR1, PYL4) and PP2C proteins (ABI1, ABI2), and hampers the inhibition of ABI1/ABI2 by ABA-bound PYR1/PYL4, thereby terminating ABA signaling. A rice ortholog of *AtABI1* and *AtABI2*, named *OsABI-LIKE2* (*OsABIL2*), is localized in both the nucleus and cytosol. OsABIL2 play an important role in regulating root development. In the presence of ABA, OsABIL2 can interact with OsPYL1, redistributing from nucleus to the cytosol and releasing the inhibition on SAPK8/10. Phosphorylated SAPK8/10 can activate downstream transcription factors to regulate the expression of ABA-responsive gene expression [60]. In this study, we showed that OsABT interacts with OsABIL2, OsPYL4, and OsPYL10 via Y2H and BiFC analyses (Figure 5). The transcriptome and qRT-PCR analyses showed that several core ABA signaling and ABA-responsive genes were obviously regulated by *OsABT*, such as *OsPYL5*, *OsPYL9*, *OsABIL2*, *OsLEA3* and *OsRAB16A* (Figure 6), indicating that *OsABT* regulates salt tolerance in rice seedling roots through the ABA signaling pathway, and is particularly important for OsABIL2-OsPYLs complexes.

Therefore, the relationship between ABA and salt tolerance deserves further attention. Xu et al. [61] found that the accumulation of ABA and the expression of the ABA synthesis gene *OsNCED3* were upregulated in *SiMYB19* transgenic rice, which showed improved salt tolerance compare with WT rice. Zhang et al. [62] found that *OsNAC45* positively regulates the ABA signaling pathway and confers salt tolerance in rice. Interestingly, *OsSAE1* negatively regulates the ABA signaling pathway and positively regulates salt tolerance in rice seedlings by inhibiting the expression of *OsABI5* [63]. Chen et al. [51] found that OsABT inhibits the ABA signaling pathway by interacting with OsABI2, and the tolerance to salt stress was higher in *OsABT* overexpressing *Arabidopsis* than in wild type plants. In this study, *OsABT* negatively regulated ABA synthesis and signaling pathways and conferred salt tolerance in rice. Salt tolerance is a complex trait involving many genes and pathways, but accumulating evidence has revealed multifaceted crosstalk between the ABA signaling pathway and the SOS signaling pathway. For example, SOS2 is able to physically interact with ABI2 and it is disrupted by the *abi2-1* mutation, which causes increased tolerance to salt shock and ABA insensitivity in plants [64]. Meanwhile, there is an inseparable connection between the ABA signaling and ROS. ABA regulates ROS production through plasma membrane and mitochondria-localized NADPH oxidases. These ROS act as important secondary messengers in regulating root growth, stomatal movement, and seed germination through the ABA signaling pathway [65]. In vitro analyses indicated that H_2_O_2_ could in turn inactivate the activities of the PP2C enzymes ABI1 and ABI2 [66,67]. In the future, it will be interesting to know whether OsABT interacts with salt stress-related proteins, especially Na^+^/K^+^ transporters, and reduces the ROS levels in rice roots by negatively regulating the ABA signaling pathway.

In conclusion, based on our experimental results and relevant literature, we proposed a working model for the *OsABT* gene to improve the salt tolerance of roots in rice seedlings (Figure 7). Salt stress causes ROS accumulation in rice roots, which seriously affects root salt tolerance in rice seedlings. *OsABT* can inhibit ROS accumulation, thereby reducing the damage caused by ROS to cells under salt stress. Moreover, *OsABT* maintains the intracellular Na^+^ and K^+^ balance by enhancing the expression of *OsSOS1* and *OsHAK5* in roots. *OsABT* also inhibits the expression of the ABA synthesis genes *OsNCED3* and *OsNCED4* and promotes the expression of the ABA catabolic gene *OsABA8ox2*, thereby inhibiting the accumulation of ABA. Meanwhile, OsABT interacts with OsPYL4/10 and OsABIL2, and negatively regulates the expression of ABA signal transduction-related genes and ABA responsive genes, resulting in the attenuation of ABA signaling. Furthermore, it was reported the ABA signaling pathway is related to ROS and SOS signaling pathway. Thus, OsABT may affect the ROS levels and Na^+^/K^+^ balance through the ABA signaling pathway to regulate root salt tolerance in rice seedlings.

## 4. Materials and Methods

### 4.1. Plant Materials and Grown Conditions

Rice (*Oryza sativa* L. cv. Nipponbare) was used as the wild-type control for physiological experiments and as a recipient for genetic transformation in this study. The *35S::OsABT* overexpression rice lines (OsABT–OE) were used as experimental materials [51]. Rice seeds were soaked in water for 2 days and germinated on a damp gauze for 2 days. The rice seeds were then transferred into 96-well plates and grown in a growth room at temperatures of 30 °C with a 14 h light/10 h dark photoperiod. After culturing in water for 3 days, 1/2 of Hoagland’s solution (pH 5.5–6.0) for 3 days, and then the full Hoagland’s solution (pH 5.5–6.0) every 3 days. For the NaCl treatment, two-week-old rice seedlings were exposed to 200 mM NaCl for 0, 12, and 24 h. The roots were quickly frozen in liquid nitrogen and stored in at −80 °C.

### 4.2. Measurements of Physiological Index

Root activity was measured using the α-naphthylamine oxidation method [68], and relative conductivity was measured using a DDS-307 conductivity meter (LeiCi, Shanghai, China). The MDA content was determined using the thiobarbituric acid method [69]. The O_2_^•−^ production rate was determined using the hydroxylamine oxidation method [70]. The H_2_O_2_ content was determined using a kit (Grace, Jiangsu, China). To measure the content of Na^+^ and K^+^, the roots were washed three times with deionized water and dried at 80 °C for 3 days. All samples were weighed and digested with 5 mL of concentrated HNO_3_. Subsequently, the sample was diluted with deionized water to 25 mL, kept for 1–2 days, and then analyzed by Inductively coupled plasma atomic emission spectroscopy (ICP-AES; ICP9000, Shimadzu, Japan) [71]. The ABA content in rice roots and leaves was measured using high-performance liquid chromatography (HPLC; Waters 2695, USA) [72].

### 4.3. RNA Extraction and qRT-PCR Analyses

Total RNA was extracted from rice roots using an RNA extraction kit (CWBIO, Jiangsu, China), and the first strand of cDNA was synthesized using a HiFiScript cDNA Synthesis Kit (CWBIO, Jiangsu, China). qRT-PCR was performed using the SYBR Premix Ex Taq^TM^ II kit (TaKaRa, Dalian, China), and the amplification reaction was performed using a CFX96TM fluorescence quantitative PCR instrument (Bio-Rad, CA, USA). Each reaction was conducted in three biological and technical replicates. The rice *Actin* gene (*Os03g0718100*) was used as an internal standard. The relative expression levels of genes were calculated using the 2^-ΔΔCT^ method. The primer sets used for the qRT-PCR are listed in Appendix A.

### 4.4. Y2H Analysis

The full-length coding sequence (CDS) of *OsABT* was fused into a pBD-GAL4 Cam vector, and the full CDSs of *OsABIL2*, *OsPYL4*, *OsPYL10* were fused with the pAD-GAL4-2.1 vector. Each construct was transformed into the yeast strain YRG2. The transformed yeast clones were first grown on the SD/-Leu/-Trp/-Ura medium and then transferred to the SD/-Leu/-Trp/-His/-Ura medium for 4 days at 30 °C. The primer sequences used for Y2H analysis are listed in Appendix A.

### 4.5. BiFC Assay

To produce a fusion with either the N- or C-terminal fragment of YFP, the full-length coding regions of *OsABT* were subcloned into pCAMBIA1300-YFP^C^ vectors, and OsABIL2, OsPYL4, and OsPYL10 were subcloned into the pCAMBIA1300-YFP^N^ vectors. The corresponding BiFC plasmids and negative controls were co-expressed in *N. benthamiana* leaves. The cells were visualized 48 h after transformation, and images were captured using a confocal laser scanning microscope (Leica SP8; Leica Microsystems, Wetzlar, Germany).

### 4.6. Transcriptome Analysis

Total RNA was isolated in triplicate from the roots of OE–3 and WT treated with 200 mM NaCl for 0 and 24 h. Total root RNA was isolated and purified using the TRIzol reagent (Invitrogen, CA, USA) to obtain mRNA. A cDNA library was established using a reverse transcriptase and DNA polymerase, and paired-end sequencing was performed using Illumina Novaseq ™6000. Significant difference analysis was performed using the DEGseq2 software. Genes with a fold difference of more than two times, that is, fold change ≥ 2 or fold change ≤ 0.5 and *p* < 0.05, were defined as differentially expressed genes (DEGs) and were annotated using the GO and KEGG databases.

### 4.7. Statistical Analysis

The averages and standard deviations (SD) of all results were calculated, and Student’s *t*-tests was performed to generate *p*-values. All analyses were performed using the SPSS 20.0 software.

## Figures and Tables

**Figure 1 ijms-23-10656-f001:**
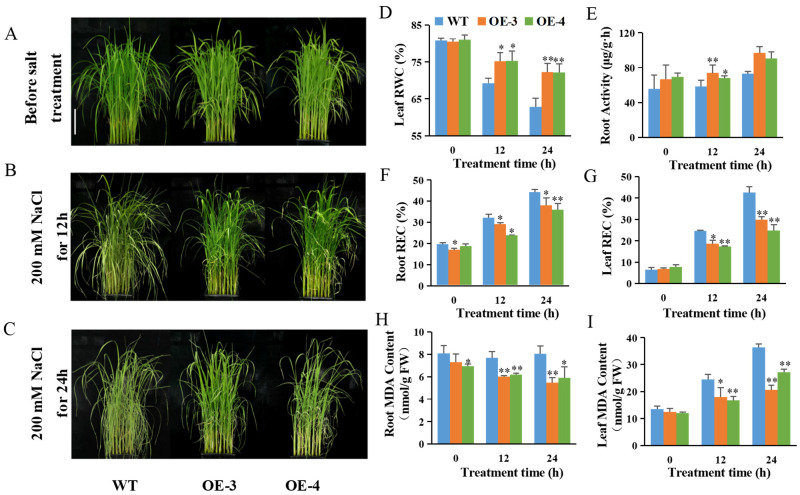
Overexpression of *OsABT* enhances the salt tolerance of rice seedlings. (**A**–**C**) Phenotypes of two-week-old OsABT–OE lines (OE–3 and OE–4) and wild-type (WT) plants under 200 mM NaCl for 0, 12 and 24 h. Scale Bar: 5 cm. (**D**–**I**) Relative leaf water content; root activity; relative root and leaf conductivity; root and leaf malondialdehyde (MDA) content of OsABT–OE lines and WT under 200 mM NaCl for 0, 12, and 24 h. Data are means ± SD of three independent experiments. Asterisks indicate significant differences compared to WT plants at * *p* < 0.05 and ** *p* < 0.01 (Student’s *t*-test).

**Figure 2 ijms-23-10656-f002:**
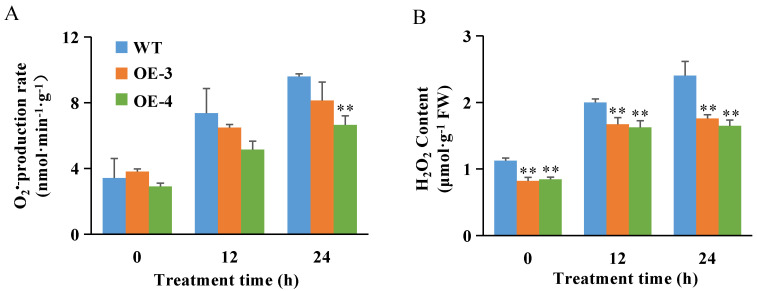
Changes of reactive oxygen species (ROS) in roots of various rice plants under salt stress. O_2_^•−^ production rate (**A**) and H_2_O_2_ content (**B**) in roots of two–week–old OsABT–OE lines (OE–3 and OE–4) and WT plants under 200 mM NaCl for 0, 12, and 24 h. Data are means ± SD of three independent experiments. Asterisks indicate significant differences compared to WT plants at ** *p* < 0.01 (Student’s *t*–test).

**Figure 3 ijms-23-10656-f003:**
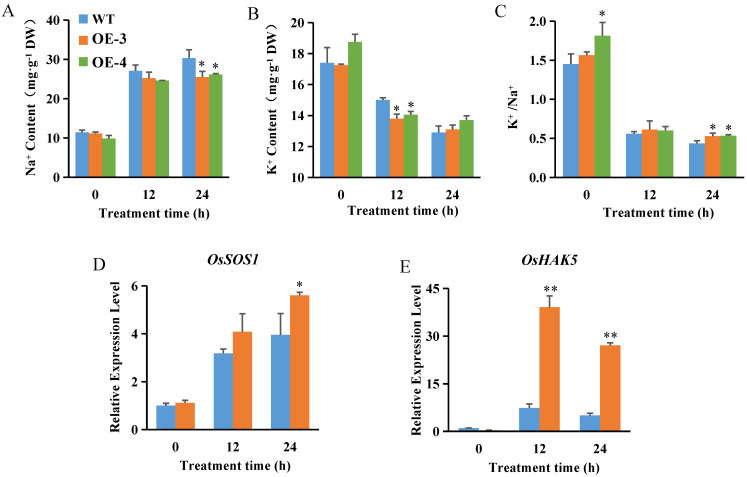
Overexpression of *OsABT* regulates Na^+^ and K^+^ levels under salt stress. (**A**–**C**) Na^+^ content, K^+^ content, and K^+^/Na^+^ ratio in the roots of two–week–old OsABT–OE lines (OE–3 and OE–4) and wild type (WT) plants exposed to salt stress (200 mM NaCl) for 0, 12, and 24 h. (**D**,**E**) Expression of *OsSOS1* and *OsHAK5* in the roots of two–week–old OsABT–OE line (OE–3) and WT plants. Root RNA was isolated and used for quantitative real–time polymerase chain reaction (qRT–PCR). *Actin* was used as an internal control. The relative expression levels were represented by fold change relative to the expression of WT before salt treatment (0 h). Data are means ± SD of three independent experiments. Asterisks indicate significant differences compared to WT plants at * *p* < 0.05 and ** *p* < 0.01 (Student’s *t*–test).

**Figure 4 ijms-23-10656-f004:**
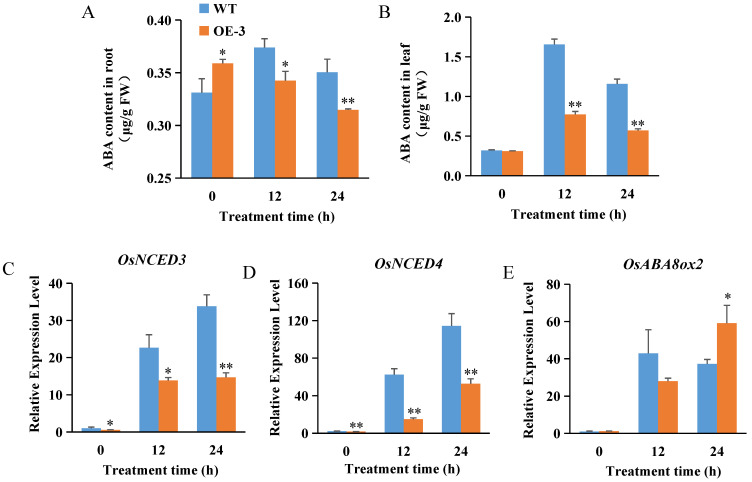
Overexpression of *OsABT* inhibits ABA synthesis under salt stress. (**A**,**B**) Root and leaf ABA content of the two–week–old OsABT–OE line (OE–3) and WT plants exposed to salt stress (200 mM NaCl) for 0, 12, and 24 h. (**C**–**E**) Root expression of the ABA synthesis genes *OsNCED3* and *OsNCED4*, and the ABA catabolic gene *OsABA8ox2* in the two–week–old OsABT–OE line (OE–3) and WT plants. Root RNA was isolated and used for qRT–PCR. *Actin* was used as an internal control. The relative expression levels were represented by fold change relative to the expression in WT plants before salt treatment (0 h). Data are means ± SD of three independent experiments. Asterisks indicate significant differences compared to WT at * *p* < 0.05 and ** *p* < 0.01 (Student’s *t*–test).

**Figure 5 ijms-23-10656-f005:**
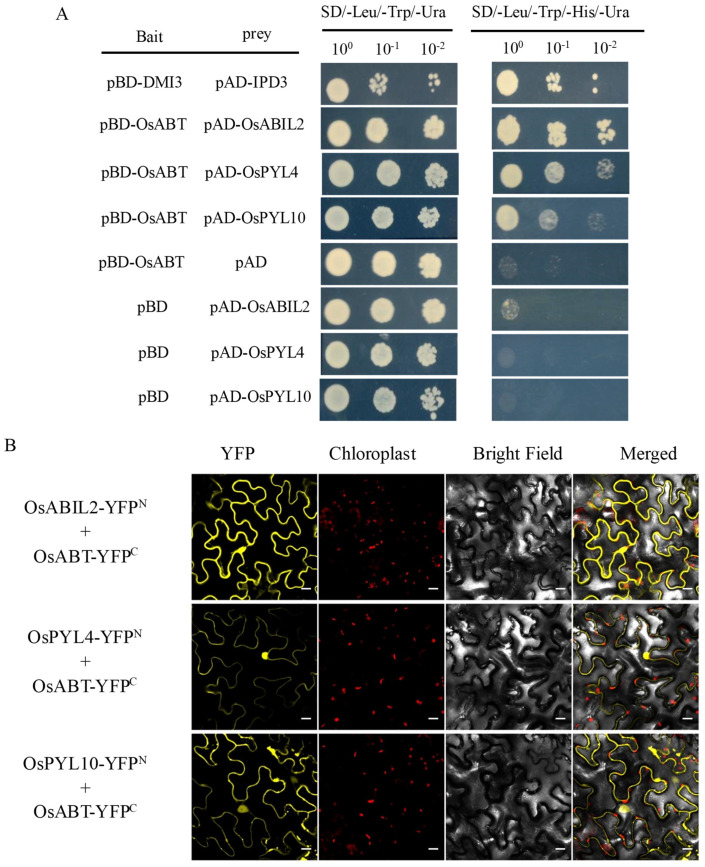
OsABT interacts with key factors of the ABA signaling pathway. (**A**) Yeast two hybrid (Y2H) assay of the interactions between OsABT and the three ABA signaling proteins OsABIL2, OsPYL4, and OsPYL10. GAL4 binding domain (BD) and GAL4 activation domain were co–transformed into yeast YRG2, and a series of 2 μL aliquots of the diluted (OD600 = 1, 0.1, 0.01) co–transformed YRG2 culture was spotted on SD/−Leu/−Trp/−Ura and SD/−Leu/−Trp/−His/−Ura media and incubated for 4 days. (**B**) Bimolecular fluorescence complementation (BiFC) assay of the interactions between OsABT and OsABIL2, OsPYL4, OsPYL10 in Nicotiana benthamiana leaves. Scale bars: 50 μm.

**Figure 6 ijms-23-10656-f006:**
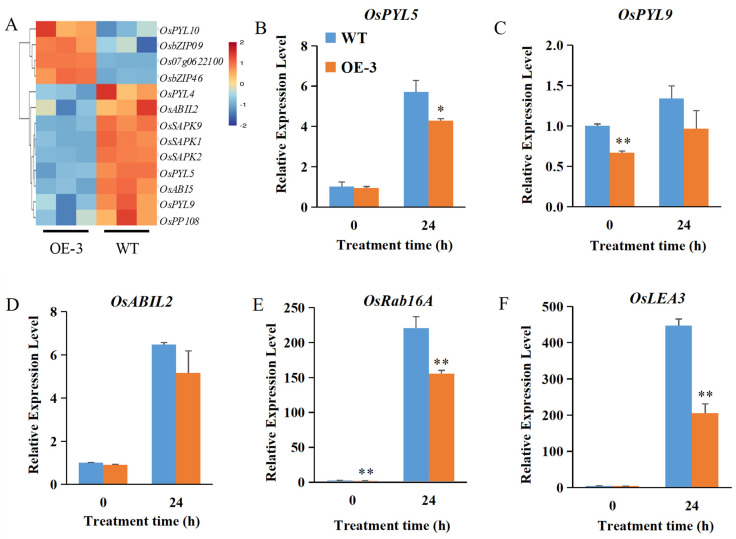
*OsABT* negatively regulates ABA signal transduction pathway. (**A**) Cluster analysis of differentially expressed genes (DEGs) in ABA signal transduction between OE–3 and WT at 24 h of salt treatment. Each column in the figure represents a sample, and each row represents a gene. Log2 processing was performed on the fragments per kilobase of exon per million mapped fragments (FPKM) value of these DEGs. (**B**–**F**) Root expression levels of ABA–related genes in 2–week–old OsABT–OE line (OE–3) and WT plants. Root RNA was isolated and used for qRT–PCR. *Actin* was used as an internal control. The relative expression levels were represented by fold change relative to the WT expression levels before salt treatment (0 h). Data are means ± SD of three independent experiments. Asterisks indicate significant differences compared to WT at * *p* < 0.05 and ** *p* < 0.01 (Student’s *t*–test).

**Figure 7 ijms-23-10656-f007:**
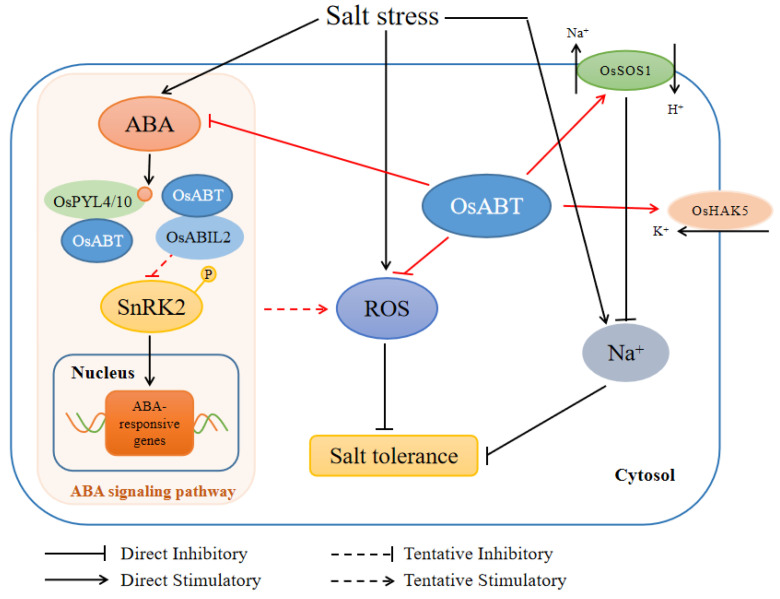
A proposed model illustrating the mechanism of the *OsABT* gene to improve the salt tolerance of roots at the rice seedling stage. *OsABT* confers salt tolerance to rice seedling roots by inhibiting ROS accumulation, maintaining intracellular Na^+^ and K^+^ balance, and regulating the ABA content. Meanwhile, OsABT interacts with OsPYL4/10 and OsABIL2, and negatively regulates the expression of ABA signal transduction-related genes and ABA responsive genes, resulting in the attenuation of ABA signaling. OsABT may affect the ROS levels and Na^+^/K^+^ balance through the ABA signaling pathway to regulate root salt tolerance in rice seedlings.

## Data Availability

Data available upon request from the corresponding author.

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
