# Peer review of "OsABT Is Involved in Abscisic Acid Signaling Pathway and Salt Tolerance of Roots at the Rice Seedling Stage"

_ijms, 2022, doi:10.3390/ijms231810656_

Round 1

Reviewer 1 Report

Using overexpression lines, the authors demonstrated straightforward that OsABT regulates rice seedling root tolerance to salt through ABA signaling pathway. The data are solid and confident as shown by the differences between the OE lines / WT in ROS, MDA, ABA contents and relative expression levels of relevant genes associated with N, K and ABA-metabolism. However, I would suggest further improvement of the manuscript before acceptance for publication with following minor revision by the authors.

1.     Line 81, please indicate the accession number of OsABT.

2.     Line 93, “T6 generation” should be “T6 generation”.

3.     Line 206-207, It seems to me the authors performed point-by-point hybridization rather than the Y2H, the authors should explain more in details.

4.     Line 252-259, more detailed description for the lowered expression of OsLEA3 and OsRAB16 is required.

5.     In the figure legends, “Student’s t test” should be “Student’s t-test”.

6.     Line 270, cited references[19-23,55-57], I don’t think it is necessary to cite a lot of references, the major ones will be enough. I would think this principle applies to the whole text for the citation.

7.     Line 311-312, “affecting the accumulation and function of ABA in leaves” is doubtful, it is unlikely associated with the function of ABA.

8.     Line 368, “14 h light/8 h dark photoperiod” looks strange?

9.     Line 374-378, please specify the methods used for measurements of root activity, MDA content, and O2•- production rate.

10.  Line 393, Please provide the accession number for the Actin gene.

11.  Line 411, Please indicate the manufacturer of the confocal microscope used in the experiment.

12.  Figure 7, indicate the meanings of different lines used.

Reviewer 2 Report

ROS and ABA interaction is well known as important in plant stress response. However, the molecular aspect of this interaction is limited. This study might give insights in the interaction of ROS and ABA during salt stress to improve rice tolerance. However, I have some comments and suggestion on the MS:

Line 206-207 “Therefore, we performed a yeast two-hybrid (Y2H) analysis to identify the interaction between OsABT the ABA receptors OsPYL4, OsPYL10, and the PP2C protein OsABIL2.” Is there any reason why only OsPYL4, OsPYL10, and OsABIL2 were tested in interaction with OsABT? Several OsPYL proteins interact with OsABIL2 including OsPYL1/3/5/6/7/8/9 (doi: 10.1186/s12284-015-0061-6). Have they been tested?

Figure 6. Transcriptome analysis commonly was done using at least two independent OE lines. Data from at least one more OE line must be included.

Line 357-358 “OsABT interacts with both PYLs and PP2C proteins to block the interaction between them, resulting in the attenuation of ABA signaling.” The data does not support this conclusion. Please clarify.

OsABIL2 localizes in cytosol (doi: 10.1186/s12284-019-0297-7), but it interacts with OsABT in cytosol and nucleus (Figure 5B). Does OsABT cause the shutter of OsABIL2 to nucleus? OsABT action or contribution on function of OsABIL2-OsPYL complex seems to be important to ABA signaling. I suggest make it clearer the mode of action of OsABT in Figure 7, especially in OsABIL2-OsPYL complex.

In the line of these comments, Discussion should be improved.

Round 2

Reviewer 2 Report

I suggest Reject and re-submit after completing the required experiments to support the authors’ conclusions:

In this study, OsABT overexpressing lines are the main plant materials. Transcriptome data (Figure 6), gene expression (Figure 4, 6), and ABA content data (Figure 4) of at least two independent primary transgenic lines is a common requirement in many plant science journals.  I strongly recommend supplementing this.

The molecular mechanism of OsABT is hard to be elucidated by gene expression solely because OsABT was not proven as a transcription regulator. Therefore, the direct regulation of OsABT on ROS and Na+/K+ balance (Figure 7) is not convinced. Instead, the interaction aspect of OsABT provides more clues as WD40 proteins are assisted in protein-protein interactions (doi.org/10.1038/s41598-021-81969-z). The MS can be improved by exploiting more interactors of OsABT to see if any among them involve ROS and Na+/K+ balance as how it works in regulating ABA.

Author Response

Thank you very much for your comment.

We have problems with single transgenic lines in terms of transcriptome sequencing, etc., and we will continue to repeat in our further research.

Fig. 7 is a model for OsABT to improve the salt tolerance of roots at the rice seedling stage, which is proposed according to the experimental results of this paper and relevant literature. The dotted line in the figure indicates the possible functional relationship according to the literature, which needs to be confirmed by further research.

In the revised version, we have enriched the discussion to make the speculation more convincing.

Round 3

Reviewer 2 Report

As in previous review round, the study is interesting but data quality needs improvement. I suggest Reject.